# Video2StyleGAN: Disentangling Local and Global Variations in a Video

## Abstract

Image editing using a pre-trained StyleGAN generator has emerged as a powerful paradigm for facial editing, providing disentangled controls over age, expression, illumination, etc. However, the approach cannot be directly adopted for video manipulations. We hypothesize that the main missing ingredient is the lack of fine-grained and disentangled control over face location, face pose, and local facial expressions. In this work, we demonstrate that such a fine-grained control is indeed achievable using pre-trained StyleGAN by simultaneously working across multiple (latent) spaces (i.e., positional, W+, and S spaces) and combining the optimization results. Building on this, we introduce Video2StyleGAN that takes a target image and driving video(s) to reenact the local and global locations and expressions from the driving video in the identity of the target image. As a result, we are able to generate high-quality videos at $1024^2$ resolution without training on video data. We evaluate the effectiveness of our method over multiple challenging scenarios and demonstrate clear improvements in terms of LPIPS over alternative approaches trained on video data (FOMM Siarohin et al. (2019), LIA Wang et al. (2022), and TPS Zhao & Zhang (2022)) and comparable scores in terms of FID, keypoint distance, and identity preservation.

## 1 Introduction

Generative modeling has seen tremendous progress in recent years, with multiple competing solutions, including generative adversarial networks (GANs) (Karras et al., 2020a; 2021a), variational autoencoders (VAEs) (Razavi et al., 2019), diffusion network (Ramesh et al., 2022), and autoregressive models (ARs) (Esser et al., 2021). In this paper, we focus on GANs and in particular, the StyleGAN architecture that produces high-resolution output. This architecture has started a wave of research exploring semantic image-editing frameworks (Shen et al., 2020; Patashnik et al., 2021; Abdal et al., 2021c). These approaches first embed a given photograph into the latent space of StyleGAN and then manipulate the image using latent space operations. Example editing operations in the context of human faces are global parametric image edits to change the pose, age, gender, or lighting, or style transfer operations to convert images to target cartoon styles. While these edits are generally successful, it is still an open challenge to obtain fine-grained control over a given face, e.g., face location in the image, head pose, and facial expression. While such fine-grained control is beneficial but optional for editing single images, they are an essential building block for creating a high-res video from a single image and other video editing applications.

We investigate the following questions: *How can we embed a given video into the StyleGAN latent space to obtain a meaningful and disentangled representation of the video in latent space? How can we create a video from a single image, mainly by transferring pose and expression information from other videos?* It is somewhat surprising how difficult it is to embed fine-grained controls into Style-GAN. Direct solutions are either over-regularized or under-regularized. Over-regularization leads to the controls being ignored so that the given reference image hardly changes; under-regularization leads to unnatural face deformations and identity loss. Our main idea is to make use of different latent spaces to encode different types of information: *positional code* controls the location of the face in the image (i.e., translation and rotation); *W space* controls global edits such as pose and some types of motion; *S space* and generator weights control local and more detailed edits of facial expressions. This hierarchical (code) structure allows the extraction of semantic information from given driving videos and their transfer to a given photograph. See Fig. 1.

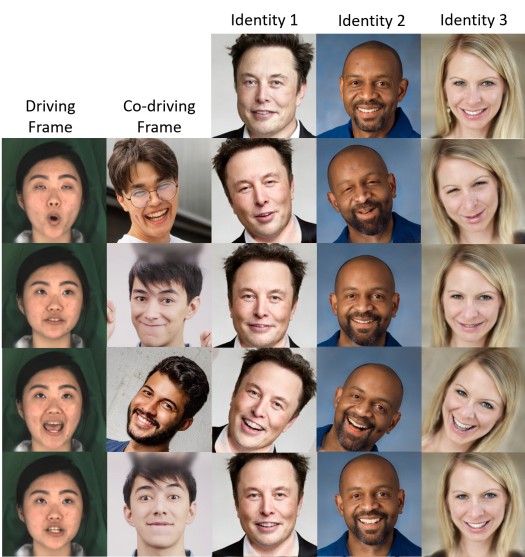

Figure 1: **Fine-grained control.** We present Video2StyleGAN, a video editing framework capable of generating videos from a single image.

We compare ours against multiple SOTA alternatives: FOMM Siarohin et al. (2019), LIA Wang et al. (2022), and TPS Zhao & Zhang (2022). Other works (Alaluf et al., 2022; Tzaban et al., 2022) use $W$ and/or $S$ spaces for video editing, but their task is different from ours. Our main contributions are: (i) proposing a facial reenactment system that uses the pre-trained StyleGAN3 to transfer the motion and local movements of a talking head. We generate temporally consistent high-res $1024^2$ video editing *without* requiring additional training on videos while the competing works demonstrate generation at $256^2$ resolution and are trained on videos.;(ii) providing insights into the $W$ and the $S$ spaces to disentangle both local and global variations in a video (e.g., fine-grained control over eye, nose, and mouth movements, in addition, to pose control) while preserving the identity of the target person. We are the first to hierarchically combine ($W+$, $S$, Fourier features, and filter weights) in a 'non-trivial' manner.; and (iii) directly extracting the local and global variations from multiple videos to reenact a given image (e.g., modify local features like eyes, nose from one video, and other global features like pose and rotation from another video). To the best of our knowledge, this is not shown in any of the previous video editing works trained on videos, let alone a network only trained on images.

## 2 RELATED WORK

**State-of-the-art GANs.** Recent improvements to the loss functions, architecture, and availability of high-quality datasets Karras et al. (2021b) have improved the generation quality and diversity of Generative adversarial Networks (GANs) (Goodfellow et al., 2014; Radford et al., 2015). Owing to these developments, Karras et al. published a sequence of architectures (Karras et al., 2017; 2021b; 2020b;a; 2021a) leading to state-of-the-art results on high quality datasets like FFHQ Karras et al. (2021b), AFHQ Choi et al. (2020), and LSUN objects Yu et al. (2015). The latent space learned by these GANs has been explored to perform various tasks such as image editing (Shen et al., 2020; Abdal et al., 2019; Patashnik et al., 2021; Abdal et al., 2021c) or unsupervised dense correspondence computation (Peebles et al., 2021). While recent 3D GANs showed promise in generating high-resolution multi-view-consistent images along with approximate 3D geometry (Chan et al., 2021; Deng et al., 2021; Or-El et al., 2021), their quality still lags behind 2D GANs. In this work, we build upon the state-of-the-art generator StyleGAN3 Karras et al. (2021a) that exhibits translation and rotation invariance with respect to the generated image.

**Image projection and editing using GANs.** There are two building blocks required for GAN-based image and video editing. First, one needs to project real images into the GAN's latent space.

In the StyleGAN domain, Image2StyleGAN Abdal et al. (2019) uses the extended $W+$ latent space to project a real image into the StyleGAN latent space using optimization. Focusing on improving the reconstruction-editing quality trade-off, methods like II2S Zhu et al. (2020b) and PIE Tewari et al. (2020b) propose additional regularizers to ensure that the optimization converges to a high-density region in the latent space. While other works (Zhu et al., 2020a; Richardson et al., 2020; Tov et al., 2021; Alaluf et al., 2021a) use encoders and identity-preserving loss functions to maintain the semantic meaning of the embedding. Recent works, PTI Roich et al. (2021) and HyperStyle Alaluf et al. (2021b) modify the generator weights via an optimization process and hyper network, respectively. Such methods improve the reconstruction quality of the projected images.

Second, latent codes need to be manipulated to achieve the desired edit. For the StyleGAN architecture, InterFaceGAN Shen et al. (2020), GANSpace Härkönen et al. (2020), StyleFlow Abdal et al. (2021c), and StyleRig Tewari et al. (2020a) propose linear and non-linear edits in the underlying $W$ and $W+$ spaces. StyleSpace Wu et al. (2020) argues that the $S$ space of StyleGAN leads to better edits. CLIP Radford et al. (2021) based image editing (Patashnik et al., 2021; Gal et al., 2021; Abdal et al., 2021a) and domain transfer (Zhu et al., 2022; Chong & Forsyth, 2021) also study the StyleGAN and CLIP latent spaces to apply StyleGAN based editing on diverse tasks. Motivated by these successes in the image domain, we now explore applications in the video domain.

**GAN-based video generation and editing.** GAN based video generation and editing methods (Menapace et al., 2021; Munoz et al., 2020; Tulyakov et al., 2018; Wang et al., 2021; Yang et al., 2023; Xu et al., 2022; Tzaban et al., 2022; Yao et al., 2021) have shown remarkable results on $128^2$, $256^2$, and $512^2$ spatial resolutions. Owing to the higher resolution and disentangled latent space of the StyleGAN, multiple works in this domain either use the pre-trained StyleGAN generator to construct a video generation framework (Fox et al., 2021; Alaluf et al., 2022; Tzaban et al., 2022) or reformulate the problem by training additional modules on top of StyleGAN and using the video data to train the networks (Skorokhodov et al., 2021; Wang et al., 2022; Tian et al., 2021; Ren et al., 2021; Yin et al., 2022). Among them is StyleVideoGAN Fox et al. (2021), which is based on the manipulation in $W+$ space of StyleGAN. Related to the pre-trained latent space based method, other methods (Alaluf et al., 2022; Tzaban et al., 2022) analyze in the $W$ and $S$ spaces of StyleGAN to edit an embedded video. These methods solve a different task than ours and instead focus on editing an embedded video in different spaces of StyleGAN. Others like StyleGAN-V Skorokhodov et al. (2021) and LIA Wang et al. (2022) retrain the modified StyleGAN architecture on videos. Note that our method is a latent space based method on StyleGAN3 trained on images that do not require additional video training. LIA is also trained on different datasets than ours and cannot control the individual components of the generated image by deriving information from different videos. In Sec. 4, we compare against the relevant works addressing the same problem as ours. *Code for Fox et al. (2021) was not available at the time of this submission.*

## 3 METHOD

### 3.1 SETUP AND NOTATIONS

Given a *reference image* $I_{\text{ref}}$ and frames of a *driving video* $D := \{D_j\}$, our goal is to produce a sequence of video frames $V := \{V_j\}$ that enacts a talking head with the identity of $I_{\text{ref}}$ and pose and expressions, both local and global, from the driving video $D$. Optionally, a *co-driving video* $CD := \{CD_j\}$ may be provided as input. Given these inputs, we develop a framework to produce a disentangled representation of a driving video, such that we can encode both its global and local properties and control them separately to produce an output video $V$.

Let $\mathcal{G}$ be the pre-trained StyleGAN3 Karras et al. (2021a) generator. For the task of reenactment of the talking head, using a single (identity) image (See Fig. 2), we consider both the $W+$ and $S$ spaces of StyleGAN3. Let $w+ \in W+$ and $s \in S$ be the variables in the respective spaces for any input image. We recall that activations in the $S$ space are derived from the $w+$ codes using $s := A(w+)$, where $A$ is an affine transformation layer in the StyleGAN3. In addition to these two latent spaces, let the first layer of the StyleGAN3 $\mathcal{G}$ producing interpretable Fourier features to be represented by $F_f$. To encode a given driving video into the latent space of StyleGAN3, we project the individual frames of the video into the latent space. We use ReStyle Alaluf et al. (2021a) to project the canonical frames of the video and the reference image (i.e., after the FFHQ-based

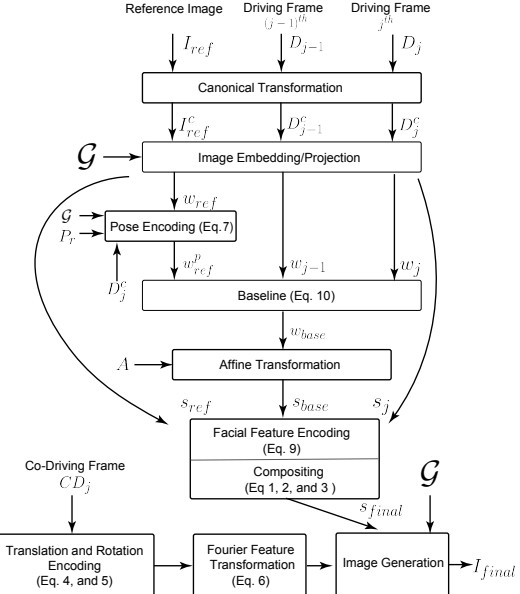

Figure 2: **Video2StyleGAN pipeline.** Flow diagram of our Video2StyleGAN method. Each box represents a local or global encoding and editing module used by our method. See Sec. 3 for details.

transformation) into the $W+$ space ($w+ \in \mathbb{R}^{18 \times 512}$) of StyleGAN3. Let the resulting *reference image* be represented by $I_{ref}^c$ and $w_{ref}$ be the corresponding $w+$ code. For the reference image, we further optimize the generator using PTI Roich et al. (2021) to improve the reconstruction quality.

### 3.2 METHOD OVERVIEW

Fully controllable and fine-grained video generation typically comes with three challenges. We first discuss these challenges and then propose the solution in Sec. 3.3. Key to our solution is a novel hierarchical embedding method that allows operating across multiple latent spaces. First, in Sec. 3.3.1, we define an algorithm to smoothly embed and transfer the transformations from a driving/co-driving ($D/CD$) video to generated frames ($V$). Second, to handle *handle 3D pose variations*, our solution uses a *masked $W+$* space with regularizers defined in Sec. 3.3.2. Third, we define solutions in Sec. 3.3.3 and Sec. 3.3.4 to transfer local and global variations. In the following subsection, we define our overall hierarchical video embedding method.

### 3.3 VIDEO2STYLEGAN METHOD

We now define and formulate the components (see Fig. 2) of our framework and describe how to extend the method to handle controls from multiple videos. A regularized combination of these components/building blocks comprises our final Video2StyleGAN method.

First, to account for the global translation in the talking head, we canonicalize the input video(s) by estimating rotation and translation parameters of the driving or a co-driving video ($D/CD$) using the Canonical Transformation (Sec. 3.3.1), and use the extracted transforms on the given image. Optionally, we can omit these changes to stay faithful to the original parameters in a given image.

Second, to achieve identity-preserving 3D pose transformation, we perform pose changes via the driving or co-driving video using pose transfer (Sec. 3.3.2). Again, we can omit such changes, i.e., use the pose of the given image without matching it to a driving frame.

Finally, we merge information from $S$ space (Sec. 3.3.3) and $W+$ space (Sec. 3.3.4) analysis to achieve fine-grained control over video generation. Specifically, we use the $S$ space to control the degree of local changes (eyes, nose, and mouth), and the $W+$ space to encode the residual motion from the driving video $D$. There are two types of regularized sets of $s$ activations that stem from

our analysis in Sec. 3.3.3. These are the activations of the reference frame $I_{ref}^c$ i.e., $s_{ref}^p \in \mathcal{X}_s$ (Sec. 3.3.3) and driving frame $D_j^c$ i.e., $s_j^p \in \mathcal{X}_{sd}$. The local edits are given by:

$$s_{local} := \alpha s_{ref}^p + \beta s_j^p. \tag{1}$$

For the $W+$ space, there are also two types of $w+$ codes; one $w_{ref}$ (See Sec. 3.1), which encodes the reference image, and another, obtained from Eq. 10 (Sec. 3.3.4), which extracts the residual motion from $D$ not captured by the $S$ space. We identify the $w+$ code layers $3-7$ (Modified Baseline) to produce the best results when combined with the $S$ space. Let $\mathcal{X}_{orig} := \{x \in \mathbb{R}^{512}\}$ be the original $w+$ encoding of the reference image $I_{ref}^c$ containing $w+$ codes of layers $3-7$. Similarly, we denote another set of $w+$ codes for these layers (Eq. 10) as $\mathcal{X}_w := \{x \in \mathbb{R}^{512}\}$. We first transform these $w+$ codes to corresponding $s$ activations.

Let $A_l$ be the affine function of layer $l$ of $\mathcal{G}$. We compute $\mathcal{X}_{origs} := \bigcup\limits_{i=3}^{7} A_l(w_l)$ and $\mathcal{X}_{ws} := \bigcup\limits_{i=3}^{7} A_l(w_l')$, where $w_l \in \mathcal{X}_{orig}$ and $w_l' \in \mathcal{X}_w$, respectively. These regularized spaces can be combined to control the intensity of local and global variations. Based on $s$ activation position in $\mathcal{G}$, we can combine as follows:

$$s_{final} := s_{local} + \gamma s_{base}^p, \tag{2}$$

where $s_{base}^p \in \mathcal{X}_{ws}$, such that it matches the $s$ activation position computed in set $\mathcal{X}_s$. For other $s$ activations:

$$s_{final} := \zeta s_{ref}^q + (1 - \zeta)s_{base}^q \tag{3}$$

where $s_{ref}^q \in \mathcal{X}_{origs}$ and $s_{base}^q \in \mathcal{X}_{ws}$. Note that $\alpha, \beta, \gamma, \zeta$ can be controlled separately to produce a desirable animation. For example, Eq. 2 can be used to enhance the motions in the eyes, nose, and mouth, and Eq. 3 can be used to include additional motions in the head from $D$. Now we define each component in detail.

### 3.3.1 Canonical Transformation

This building block solves the first challenge: given a sequence of positions of the talking head from a driving video $D/CD$, how could one transfer this information smoothly to a reference image $I_{ref}^c$ to produce a sequence? We exploit the translation and rotation invariance property of the StyleGAN3 architecture to encode the rotation and translation of the talking head. We recall that the Fourier features of StyleGAN3 Karras et al. (2021a) can be transformed to produce an equivalent effect on the output image. We define a tuple $(t_x, t_y, r)$, where $t_x$ and $t_y$ are the horizontal and vertical translation parameters, and $r$ is the rotation angle. First, in order to determine the translation and rotation changes from the canonical positions present in FFHQ Karras et al. (2019), we use a state-of-the-art landmark detector ageitgey (2018) on each frame of the video to determine the frame-specific $(t_x, t_y, r)$ parameters. For each frame, we compute a vector connecting the average of the positions of the *eye* landmarks and the *mouth* landmarks. We use them to compute the relative angle between the canonical vertical vector and the current face orientation that we use to encode the rotation of the head. Let $e_l$ and $e_r$ be the eye landmarks (left and right, resp.) and $m_l$ be the mouth landmarks predicted by the landmark detector $L_d$. Then,

$$\vec{e} := 0.5(\mathbb{E}(e_l) + \mathbb{E}(e_r)) \quad \text{and} \quad \vec{v} := \mathbb{E}(m_l) - \vec{e}$$

and

$$r := d_{cos}(\vec{u}, \vec{v}), \tag{4}$$

where $\mathbb{E}$ denotes average, $d_{cos}$ is the cosine similarity function, and $\vec{u}$ is the up vector. Similarly, as per the FFHQ transformation, the translation parameters are given by,

$$\vec{t} := \vec{e} - \vec{e}', \tag{5}$$

where $\vec{e}'$ is the midpoint of the canonical FFHQ transformed image, and $\vec{t}$ is a column vector representing $t_x$ and $t_y$. The transformations on the Fourier features $F_f$ to produce the desired rotation and translation effects on a given image are given by,

$$F_f'(t_x, t_y, r) := F_f(\tau(t_x, t_y, r)) \tag{6}$$

where $\tau$ represents the transformation (see Fig. 5 in Supplementary Materials).

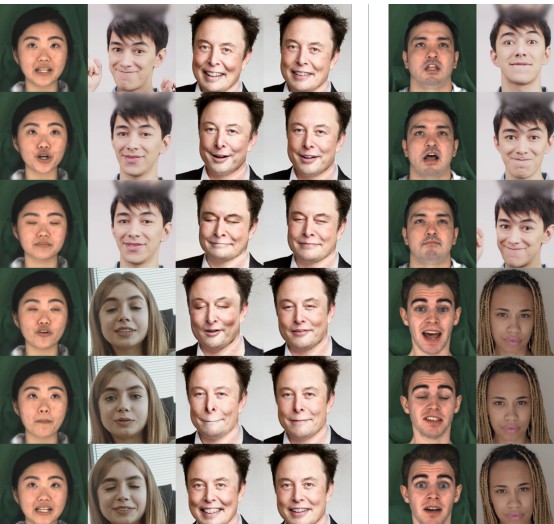

Figure 3: **Comparison with the baseline.** In each sub-figure, the first column shows the driving frames, the second column shows the co-driving frames, the third column shows the results of the baseline method, and the last column shows our results. Please see the supplementary video.

### 3.3.2 GLOBAL POSE ENCODING

Consistent with previous works Abdal et al. (2021c), we first use the first two layers StyleGAN to transfer the pose. We observe that applying this change stretches the face area and the eyes, however, the mouth and nose positions remain unchanged making the output face unrealistic. While a less constrained approach of transferring the first eight layers makes a plausible pose change at the cost of identity loss (see supplementary video).

We now propose our optimization setup to match the pose information. Specifically, we set up an objective to optimize for the pose (i.e., yaw, pitch, and roll) of a given image to match the pose of the driving video. We consider optimizing two objectives on a masked $W+$ space of the Style-GAN3, i.e., pose matching and identity preservation. For pose matching, we use a pose regression model cunjian (2019) which, given a valid frame of video, outputs yaw, pitch, and roll. To ensure identity preservation, we apply an additional $L1$ regularization to the masked $W+$ space to restrict the optimized latent to the initial latent code. We apply our optimization on the first $8$ layers. Another challenge is to perform this optimization on real images embedded using PTI Roich et al. (2021). In this case, optimizing the latent code directly creates severe artifacts. Hence we apply this optimization to a more semantically meaningful original generator latent space and then transfer the PTI-trained generator weights on top for the details. We found that this technique works best in the projected real images case (see Supplementary video). The final optimization is given by:

$$w_{ref}^p := \underset{w_{ref}^{1:8}}{\arg\min} \underbrace{L_{mse}(P_r(G(w_{ref})), P_r(D_j))}_{\text{pose matching}} + \underbrace{L_1(w_{ref}, w_{ref}^p)}_{\text{identity preservation}}, \quad (7)$$

where $w_{ref}$ is the w code for $I_{ref}^c$ and $w_{ref}^{1:8}$ is the masked $w+$ code for the first eight layers of Style-GAN3, $L_{mse}$ represents the $MSE$ loss, and $P_r$ is the output of the pose regression model cunjian (2019).

In Fig. 6 (Supplementary Materials), we show the results of the pose matching from a random frame in the driving video. The figure shows different results of pose changes made to the reference images under a given pose scenario in the driving video.

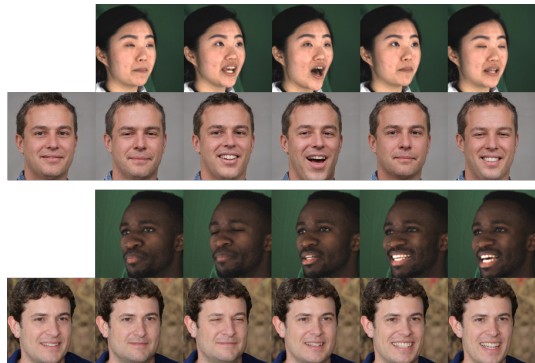

Figure 4: **Fine-grained local control.** Local information transfer without the global changes like pose. In each sub-figure, the top row represents driving frames and the bottom row shows a reference image and local edits.

### 3.3.3  LOCAL FACIAL FEATURE ENCODING

In order to automatically identify the feature maps and the corresponding $s \in S$ parameters responsible for affecting the motion of the semantic regions. Note that we do not use gradient maps Wu et al. (2020) in our analysis and apply a more fine-grained approach on activation maps based on results of the previous works Abdal et al. (2021b). Essentially, we match the activations in these layers with semantic segmentation regions obtained using a segmentation network. We use a semantic part segmentation network, BiSeNet Yu et al. (2018), trained on the CELEBA-HQ Karras et al. (2017) dataset, to determine such layers. First, given a set of images and their feature maps extracted from the StyleGAN3, we first compute the segmentation map of the image using BiSeNet. Second, we compute the normalized maps using $\min - \max$ normalization per feature channel of the feature maps. Third, to match the spatial size of these masks, we upsample these masks to match the spatial size of the target mask using bilinear interpolation. In order to convert these normalized features into hard masks, we threshold these maps to be binary. Finally, we compute the $IOU$ scores of the three semantic components derived from the set of images by comparing them with these binary masks.

Let $SegNet$ be the semantic part segmentation network (e.g., BiSeNet), $M_{fg}$ be the semantic component in consideration, $M_{bg}$ be other semantic components including background given by $SegNet(I_{ref}^c)$. Let $C_l$ be the feature map at layer $l$ of StyleGAN3 after applying the $\min - \max$ normalization, upsampling, and binarization to the map, to produce,

$$
\begin{aligned}
IOU^+ &:= IOU(M_{fg}, SegNet(C_l)) \quad \text{and} \\
IOU^- &:= IOU(M_{bg}, SegNet(C_l)).
\end{aligned}
\tag{8}
$$

Based on both the positive $IOU^+$ (eye, nose, and mouth) and negative $IOU^-$ (background and components excluding the given semantic part) $IOU$-s, we select a subset of these maps ($\mathcal{X}_m := \{x \in \mathbb{R}^{1024^2}\}$) and the corresponding $s$ parameters ($\mathcal{X}_s := \{x \in \mathbb{R}\}$) based on thresholding to be our local model for the manipulation of the semantic parts. Thus,

$$
C_l \in \mathcal{X}_m, \text{ if } IOU^+ \geq t_{fg} \text{ and } IOU^- \geq t_{bg}
\tag{9}
$$

where $t_{fg}$ and $t_{bg}$ are the thresholds. Note that $\mathcal{X}_s \subset S$. In Fig. 7 (Supplementary Materials), we show some examples of the extracted feature maps in $\mathcal{X}_m$ focusing on only a specific semantic part of the face.

### 3.3.4  RESIDUAL MOTION ENCODING

Finally, in our experiments, we found that it is sufficient to simply encode the above global and local components to perform realistic video editing using the StyleGAN3 generator. We further observe that even though the $w+$ code of the projected driving video can encode non-semantic components, which cannot be directly used for video editing, it carries other important information that is lost when shifting to the $S$ space analysis described above. Hence, the $w+$ code, despite having some undesirable effects, captures some additional semantics essential for making the motion of the

face consistent with the driving video. It is able to encode non-local effects such as stretching and squeezing of the cheek during movements in the mouth, eye regions, and chin. Only a local analysis cannot capture such coupling between the (semantic) parts. Specifically, in Fig. 3, we compute the difference vectors in the consecutive frames of the driving video and apply these transformations to the given latent representing a given image. Thus,

$$w_{base} := w_{ref}^p + (w_{j-1} - w_j) \tag{10}$$

where $w_{j-1}$ is the $w+$ code corresponding to $D_{j-1}$ and $w_j$ is the $w+$ code corresponding to $D_j$ of the driving video. Note that in Sec. 3.3, we resort to a constrained $W+$ space to apply these edits and avoid the non-semantic artifacts.

To show the artifacts and the loss of the person's identity using such a naive technique for video editing see the supplementary video. Since the previous methods (Abdal et al., 2019; Alaluf et al., 2022; Tzaban et al., 2022) use such editing in their video processing frameworks, we regard this as a baseline for our method (see Fig. 3 for a comparison).

## 4 RESULTS

### 4.1 METRICS

We use four metrics to evaluate the keypoints, identity preservation, and the quality of the frames in the resulting video. We also check the consistency of these metrics on the resulting videos (Sec. 4.4) by encoding a reverse driving video. These metrics are: **Keypoint distance** ($\Delta K$), **Identity distance** ($ID$), **LPIPS** ($LP$), and **Fréchet Inception Distance**. A description of these metrics is provided in the supplementary materials.

### 4.2 BASELINE

As mentioned in Sec. 3.3.4, we resort to Eq. 10 as a method to make consecutive edits to the $w+$ code of the embedded video which forms our baseline. Note that this method is widely used by GAN-based image editing methods like InterfaceGAN Shen et al. (2020) and GANSpace Härkönen et al. (2020). More specifically, current video editing works Alaluf et al. (2022); Tzaban et al. (2022) use the videos embedded in the $W+$ space and/or weights of the generator Roich et al. (2021) to do editing. We apply the same approach to modify a single image and generate a video using the driving and the co-driving frames. In Fig. 3, the third column in each sub-figure shows the result of the baseline method on two different identities.

### 4.3 QUALITATIVE COMPARISON

In order to visualize the quality of the resulting video, in Fig. 1, we show the results of our Video2StyleGAN method on different identities. Note that here we first match the pose of the given identity image to a driving frame and then we apply the local and global edits including the rotation and translation derived from a co-driving video. Notice the quality of the identity preservation across different editing scenarios. To compare our method with the baseline, in Fig. 3, we show the results of the editing and transformations. For embedding a real image, we use the Restyle method to produce an embedding and further optimize the generator using PTI Roich et al. (2021) by initializing with the computed Restyle embedding. Notice that the baseline approach tends to change different features like skin color and produces noticeable artifacts. In comparison, our method is able to preserve the identity of the person and successfully transfer the edits from the driving and the co-driving video. In order to show that our method works when the pose of the reference image does not match the driving frame, in Fig. 4, we show the transfer of the local information from the driving frames to a reference image. Notice the quality of edits and identity preservation in these cases. Please refer to the supplementary video.

### 4.4 QUANTITATIVE COMPARISON

In order to compute the metrics on the generated frames of our method, baseline method and other alternative techniques: FOMM Siarohin et al. (2019), LIA Wang et al. (2022), and TPS Zhao &

Table 1: **Perceptual and identity evaluation.** The best score is in **bold** and the second best is underlined.

| Method | $LP^f$ | $LP^r$ | $ID^f$ | $ID^r$ |
|---|---|---|---|---|
| Baseline | 0.423 | 0.408 | 0.58 | 0.54 |
| FOMM Siarohin et al. (2019) | 0.502 | 0.486 | 0.27 | 0.29 |
| LIA Wang et al. (2022) | 0.474 | 0.461 | **0.25** | **0.25** |
| TPS Zhao & Zhang (2022) | 0.482 | 0.467 | 0.27 | 0.28 |
| Fox et. al Fox et al. (2021) | - | - | - | - |
| Ours | **0.265** | **0.223** | 0.31 | 0.30 |

Table 2: **Keypoint distance and FID evaluation.** The best score is in **bold** and the second best is underlined.

| Method | $\Delta K^f(1e^{-3})$ | $\Delta K^r(1e^{-3})$ | FID |
|---|---|---|---|
| Baseline | 7.15 | 6.20 | 28.84 |
| FOMM Siarohin et al. (2019) | 5.03 | 4.30 | 13.37 |
| LIA Wang et al. (2022) | **4.83** | **4.21** | 8.33 |
| TPS Zhao & Zhang (2022) | 4.94 | **4.21** | 13.83 |
| Fox et. al Fox et al. (2021) | - | - | - |
| Ours | 4.91 | 5.18 | 12.91 |

Zhang (2022), we use 5 identities ($1024^2$) to produce videos of 114 frames using MEAD Wang et al. (2020) dataset. To test the consistency of the methods, in addition to computing the edits in the forward direction, we reverse the driving video and compute the edits using this reverse driving video. A consistent method should produce similar edits starting from a reference image, such that the identity, keypoints, and quality of the edits are preserved.

First, in Table 1, we compute the two metrics $LP$, and $ID$ using both the driving as well as the reverse driving video. With our generated videos supporting $1024^2$ resolution and other techniques producing videos at $256^2$, note that the $LP$ score is lower in our case for both scenarios. Our identity scores are comparable to other techniques and beat the baseline by a larger margin.

Second, in Table 2, the Keypoint Distance ($\Delta K$) of our method beats the baseline method in both scenarios showing that our method is both better at matching the keypoints as well as consistent across the driving video direction. While we do not expect to beat other methods in this metric as these methods are themselves keypoint based and our method does not need such data. Still, our method reaches very near to the scores of other methods.

Finally, to compute the quality and consistency of the edits, we measure the FID score between the frames produced by a driving video and its reverse version. The table shows that our results are comparable to the alternate techniques. This indicates that our method can produce consistent quality images across different identities and driving video scenarios similar to alternate techniques. Interestingly, we are second best when it comes to keypoint distance and FID.

## 5 CONCLUSIONS

We introduced a framework for fine-grained control for manipulating a single image using the Style-GAN3 generator. In particular, the framework is useful to edit a single image given a driving video without needing the video data for training. This problem is very challenging because existing methods either strongly overfit or underfit the driving video. We proposed a hierarchical embedding method to encode the video into the StyleGAN3 latent space. We proposed a non-trivial combination of regularized $W+$, $S$, and Fourier Feature $F_f$ spaces to achieve fine-grained control over video generation. Contrary to the previous works we can generate videos at $1024^2$ (versus $256^2$), and our method can control different components of the video separately including supporting multiple driving video inputs not seen in the previous works. Our experiments yield qualitative results in the accompanying video and quantitative results using four different metrics to demonstrate clear improvements in LPIPS scores against the state-of-the-art methods and comparable results in other metrics.

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

## A  MORE ON CANONICAL TRANSFORMATION

The transformations Alaluf et al. (2022) discussed in the main paper are not smooth across the frames. Hence, to smoothen out the anomalies, we apply a convolution operation to this sequence of parameters across the time domain. Empirically, we found a mean filter with a kernel size of 3 or

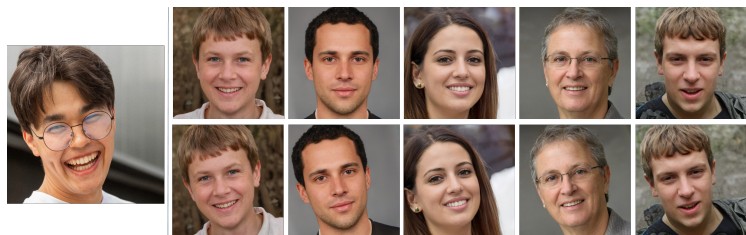

Figure 5: **Canonical transformation.** Given a driving video (left) with rotation and translation of a driving frame, our framework can transfer this information to a new reference image. Top row: reference images. Bottom row: transformed images.

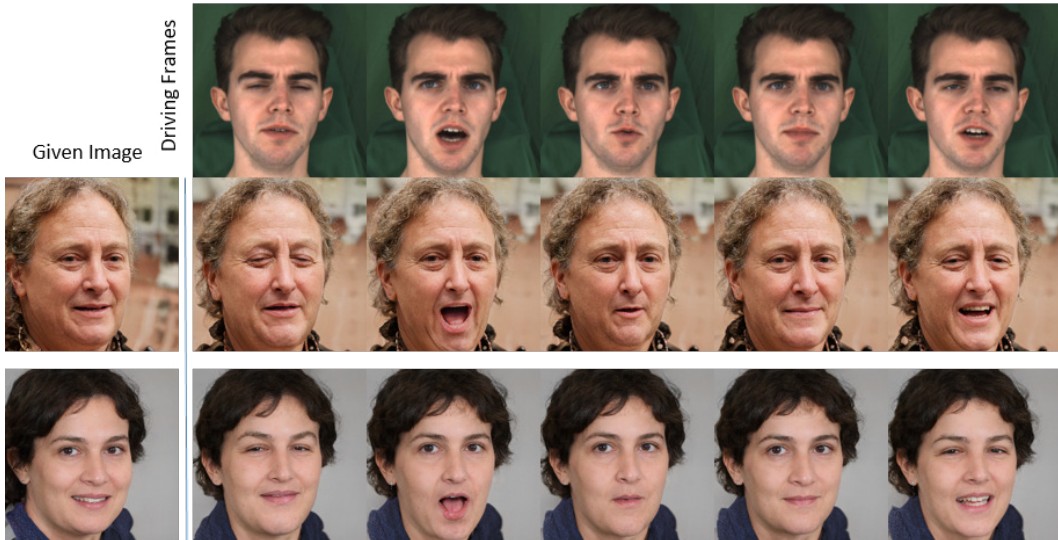

Figure 6: **Editing global pose and expressions.** Given a reference image and a driving frame, we match the pose and expressions. Each row shows an identity (reference image) and edits corresponding to the top row (driving frame).

higher to produce a smooth consistent video after the transformations are applied. Note that these parameters can be derived from co-driving video $CD$, and applied to a given image without affecting the identity. For example, we could apply the steps mentioned in Sec. 3.3 from a first driving video and apply the rotation and translation effects from a co-driving video $CD$.

## B    MORE ON GLOBAL POSE ENCODING

To solve the challenge of producing identity-preserving pose changes consistent with the driving video, we resort to analysis in the latent space of StyleGAN3. Particularly, in the context of Style-GAN, pose changes are largely associated with adding new details — stretching, squeezing, and transforming the eyes and mouth views to a target position. For this reason, we use the $W+$ space of StyleGAN3 to encode such global information. Based on the semantic understanding of the latent space by previous works Abdal et al. (2021c), we restrict encoding the pose information in the first

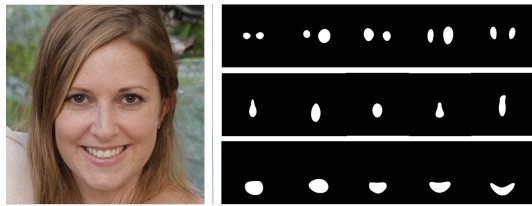

Figure 7: **Extracting local facial features.** A given image and some normalized feature maps extracted by the Local Facial Feature Encoding of our framework (see Eq. 9). The top row shows maps focusing on the eyes, the middle row shows maps focusing on the nose, and the bottom row shows images focusing on the mouth areas. We restrict latent space optimization to discovered channels responsible for the corresponding edits.

8 (out of 16) layers of the StyleGAN3 latent space which preserves both the identity and the later style layers changing global styles.

## C   MORE ON LOCAL FACIAL FEATURE ENCODING

The challenge is to encode the local information of the semantic parts, namely *eyes*, *mouth*, and *nose*, which are responsible for local changes in a talking head driving video. Note that we consider that the expression changes result in coupled variations across these semantic regions. We encode such local variations using our analysis in the $S$ space of the StyleGAN3 architecture. First, for the disentangled nature of the $S$ space in StyleGAN3 please refer to the supplementary video showing the local properties of $S$ space, which can be edited. Just for demonstration, we manually change the $S$ parameter of a given channel of a given layer of StyleGAN3 to observe the desired effect on the final image.

## D   LIMITATIONS

Although our method produces fine-grained video editing, there are some limitations of the method that stem from the nature of the StyleGAN architecture. First, our algorithm currently only considers face models. Being able to handle complete human bodies would require further extensions. Second, since there is a projection step involved in the reconstruction of a reference frame, the projection may be poorly able to model some non-semantic components like a head scarf, nose ring, etc. Third, due to the distribution of the FFHQ dataset, under large motion and extreme pose change conditions, the results might break. In future work, we would like to further explore and improve video editing by combining our analysis with other generative models, such as auto-regressive transformers and diffusion. We also propose text-driven video editing as a possible direction for future work.

## E   METRICS

1. **Keypoint distance** ($\Delta K$)**:** To measure the target edits made to the resulting video, we use the mean of L2 distance between the keypoints of the consecutive frames of the driving video and the resulting video. The keypoint detector cunjian (2019) predicts 68 keypoints given an image. We average the errors for these 68 keypoints.

2. **Identity distance** ($ID$)**:** We use a state-of-the-art Face recognition model ageitgey (2018) to compute the facial embeddings of the given reference image and the frames of the resulting video. We compute the $L2$ norm of these embeddings and take an average across all frames of the video.

3. **LPIPS** ($LP$)**:** LPIPS Zhang et al. (2018) is used to compute the perceptual distance between the two images. We use this loss to compute the similarity of the frames of the generated video to the original frame.

4. **Fréchet Inception Distance** (FID): FID Heusel et al. (2017) is used to measure the distance between a given distribution of images to a generated one. In the context of video editing, we use FID as a metric that computes the quality and consistency of the edits made to the given reference frames. This measures how much the distribution of the resulting frames changes under different edits (e.g., by reversing the driving video).

## F   TRAINING AND IMPLEMENTATION DETAILS

We use an Nvidia A100 GPU for the experiments. We use the R-Config model of StyleGAN3 for inference. Starting from a pose in the reference frame, pose encoding takes under 1 minute to converge. We set the yaw, pitch, and roll loss weights to 2 and the identity preservation loss weight to 0.04. As a default setting, we set $\alpha = -1$, $\beta = 1$, $\gamma = 1$, and $\zeta = 0.5$. Given the driving video, our method generates at the speed of $\sim$1.5 frames/sec at $1024^2$ resolution.

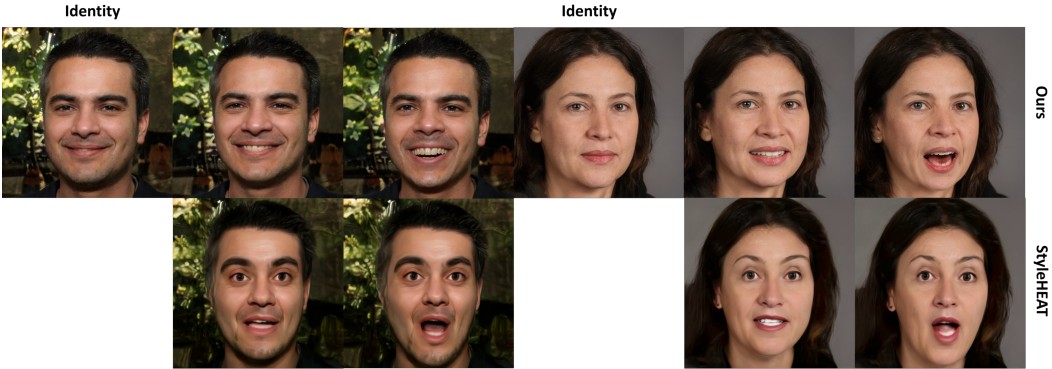

Figure 8: **Ours vs StyleHEAT.** We show the results of identity preservation using our method versus the StyleHEAT method. Notice that our method preserves the identity of the person better and at the same time preserves the details like teeth, high-frequency details on the skin, hair, etc.

## G ETHICAL CONCERNS AND BROADER IMPACT

While the work highlights the advantages of StyleGAN-based imagery for facial reenactment systems with applications in video processing/video conferencing, there are some ethical concerns that need to be checked. Particularly, such systems like DeepFakes can be misused when applied in sensitive situations or can be used to propagate misinformation. We advise caution against using the system with the intent to harm a person's autonomy, dignity, and privacy. We would encourage researchers to build systems for detecting such cases and mitigating the risks of misinformation.

## H VISUAL COMPARISON WITH STATE-OF-THE-ART METHODS

In Fig. 10 and Fig. 11, we show the stills from the videos generated by the baseline, FOMM Siarohin et al. (2019), LIA Wang et al. (2022), TPS Zhao & Zhang (2022) and our method. The reference images are shown in Fig. 12 for assessing identity preservation of different methods. Notice that our videos are generated at a higher ($1024^2$) resolution with better or similar motion/expression transfer quality as other methods. Also notice the quality of the preservation of high-quality details like teeth, hair, etc. These features are poorly represented by other methods. Zoom in to see the high-quality details of our method. See Fig. 13 and Fig. 14 for inspection.

## I COMPARISON WITH STYLEHEAT YIN ET AL. (2022)

StyleHEAT Yin et al. (2022) and Pirenderer Ren et al. (2021) are methods trained on videos and they use 3DMM parameter regressors in their frameworks. Both are StyleGAN2-based methods. We already compare with a StyleGAN2-based method, LIA Wang et al. (2022), in the main paper. Here, we show additional identity preservation and detail preserving results of our method versus StyleHEAT (StyleGAN2-based) in Figure 8. Quantitatively, in Table 3, we show our method's LPIPS score and identity preservation score versus StyleHEAT. Additionally, we also compute expression transfer performance serengil (2023) and self-reconstruction results. Self-reconstruction of a video

Table 3: **Comparison with StyleHEAT.** $L_p$: LPIPS score, $Id$: Identity Score, $E_e$: Expression conf. error, $E_p$: Expression preference, SR: Self Reconstruction

| Method | $L_p \downarrow$ | $Id \downarrow$ | $E_e \downarrow$ | $E_p \uparrow$ |
|---|---|---|---|---|
| Ours | **0.27** | **0.31** | **0.55** | **0.54** |
| StyleHEAT | 0.51 | 0.52 | 0.60 | 0.46 |
| Ours (SR) | **0.38** | **0.29** | **0.50** | **0.53** |
| StyleHEAT (SR) | 0.49 | 0.87 | 0.58 | 0.47 |

using our method depends on the PTI reconstruction (upper bound) of the first frame in a driving video. We show the results of self-reconstruction on 200 frames on 2 identities in Table 3. Note that our results are better. The Pirenderer method is worse than StyleHEAT (Table 2, row 4 in the StyleHEAT paper). Video results are shown on the supplementary webpage.

## J    DIFFERENCE TO VIDEO2STYLEGAN: ENCODING VIDEO IN LATENT SPACE FOR MANIPULATION YU ET AL. (2022)

Video2StyleGAN Yu et al. (2022), on arXiv, is different from ours. It is a concurrent work, similar to Stitch-in-time Tzaban et al. (2022) and Third Time's a Charm Alaluf et al. (2022), that deals with video editing on existing video and not face-reenactment. The method uses facial landmarks and a 3D face mesh on videos. Their code is not available to the best of our knowledge.

## K    TEMPORAL METRIC FVD.

In Table 4, we qualitatively compare against MRAA Siarohin et al. (2021) and FOMM. Ours performs a bit worse than MRAA but does *not* require any video-specific training. Additionally, we also computed the $ID$ metric for MRAA: 0.41 vs Ours: **0.31** (lower the better) which again shows that our method produces a better trade-off between reconstruction quality and motion transfer.

Table 4:  **FVD** ($1e^3$) comparison with other methods.

|  | Baseline | FOMM | LIA | TPS | MRAA | Ours |
|---|---|---|---|---|---|---|
| Trained on videos | N/A | 1.34 | 1.28 | 1.21 | **1.08** | N/A |
| Not trained on videos | 1.79 | N/A | N/A | N/A | N/A | 1.15 |

## L    ABLATION: QUANTITATIVE

In Table 5 we ablate our design choices along multiple axes. Our full method achieves a good balance between image details, temporal smoothness, and identity preservation.

Table 5:  **Ablation.** $LF$: Local Facial Feature Encoding

|  | FVD ($1e^3$) | $\Delta K^f (1e^{-3})$ | $ID$ |
|---|---|---|---|
| Baseline | 1.79 | 7.15 | 0.58 |
| $LF$ | 1.17 | 5.11 | **0.26** |
| Baseline + $LF$ | 1.38 | 5.58 | 0.43 |
| Ours | **1.15** | **4.91** | 0.31 |

## M    COMPARISON AGAINST MRAA SIAROHIN ET AL. (2021)

We provide additional quantitative (Table 4) and qualitative results in Figure 9). Ours produces higher fidelity videos with better identity preservation.

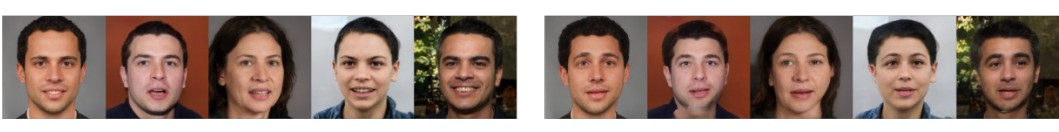

Ours                                          MRAA

Figure 9:  **Ours vs MRAA.**

## N   USER STUDY

We conducted a user study with **28** subjects, comparing ours with the methods mentioned in the main paper. Consistent with our experiments in the paper, on average, our Reconstruction/Identity preservation was preferred **62.55%** of the time, and motion/expressions (comparable to other methods) **53.05%** of the time.

## O   VIDEO RESULTS

In order to show the video results, we attach a static web page with the video results of our method. We show the ablation of the components used in our framework. Notice that adding each component improves the quality of video encoding and transfer. Also, notice that each of the building blocks can be separately controlled, unlike the previous works. To show the quality of face reenactment achieved by our method and compare it with the baseline, FOMM Siarohin et al. (2019), LIA Wang et al. (2022), and TPS Zhao & Zhang (2022) methods, we also show the comparison videos in the supplementary web-page.

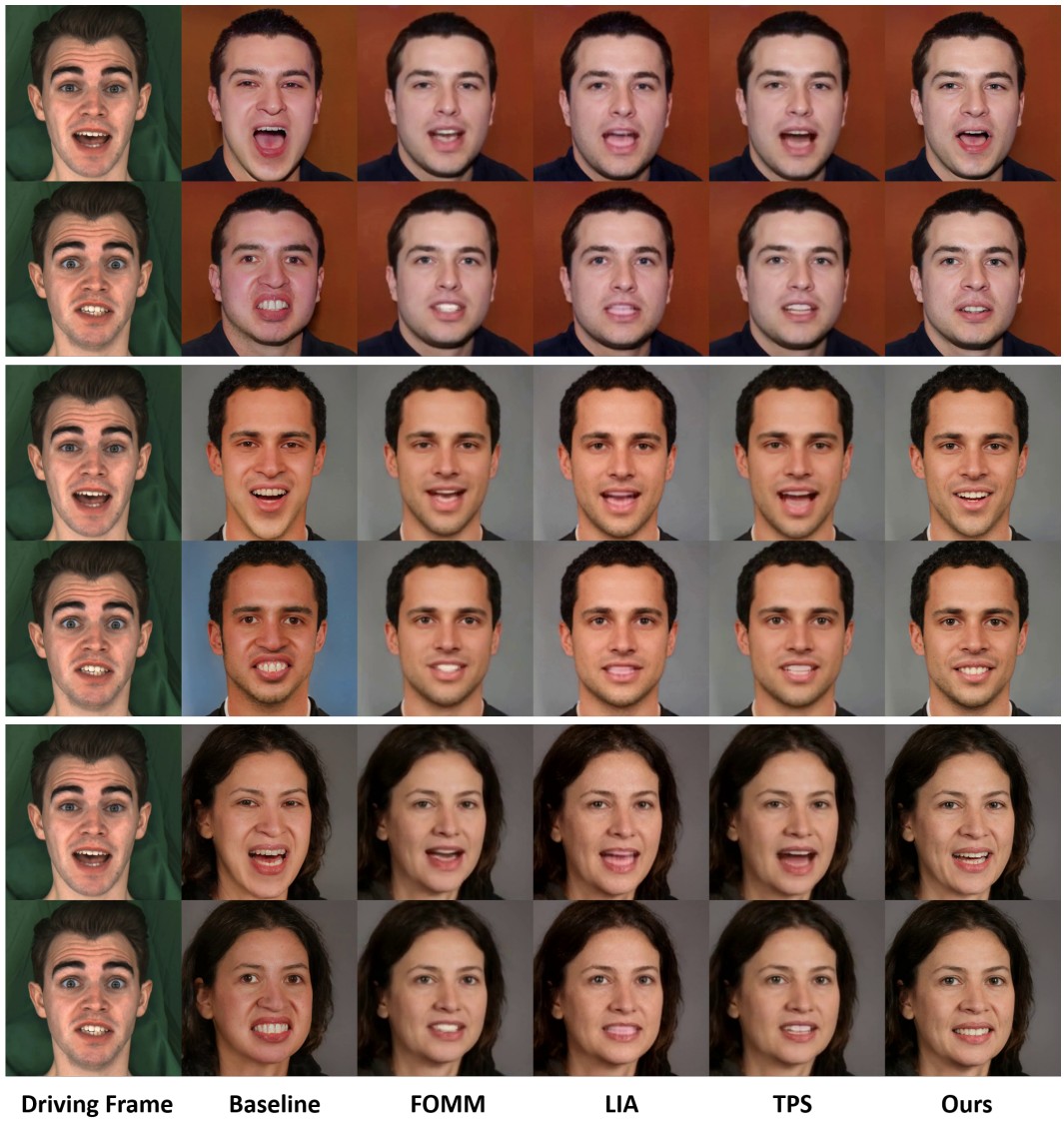

Figure 10: **Comparison with state-of-the-art methods.** Comparison of our method with the baseline, FOMM Siarohin et al. (2019), LIA Wang et al. (2022), and TPS Zhao & Zhang (2022). Notice the blurry results in other methods with poor high-frequency details e.g. teeth, hair etc. Our method preserves the details of the reference image (Fig. 12). Zoom in for details.

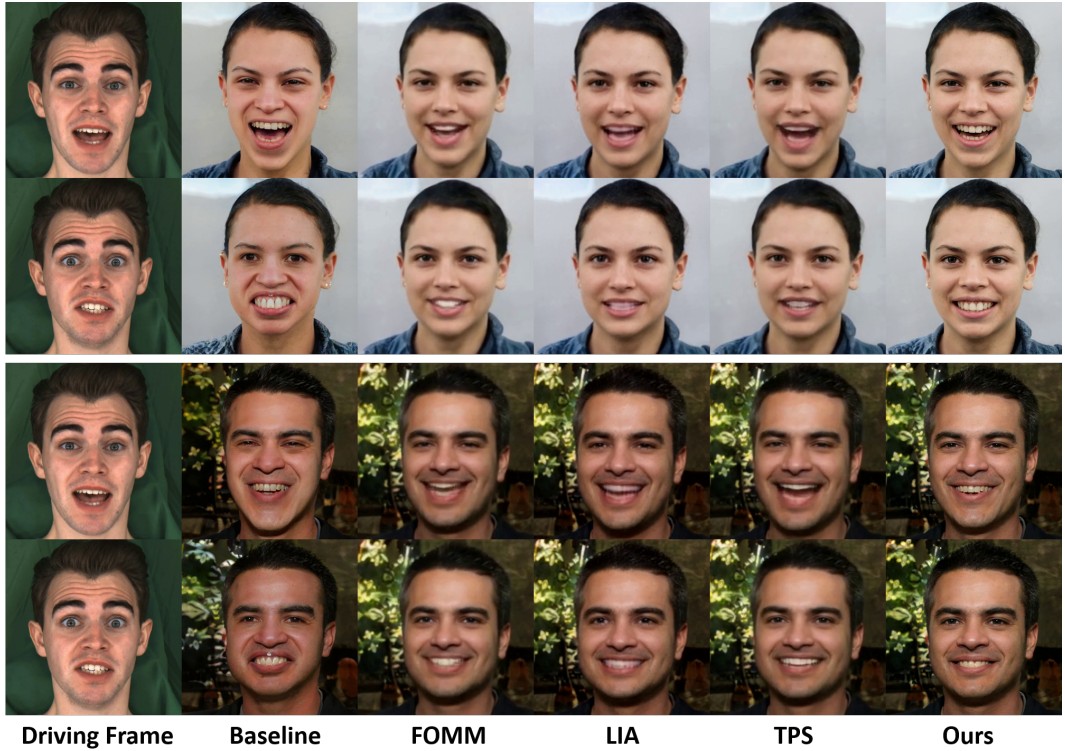

Figure 11: **Comparison with state-of-the-art methods.** Comparison of our method with the baseline, FOMM Siarohin et al. (2019), LIA Wang et al. (2022), and TPS Zhao & Zhang (2022). Notice the blurry results in other methods with poor high-frequency details e.g. teeth, hair etc. Our method preserves the details of the reference image (Fig. 12). Zoom in for details.

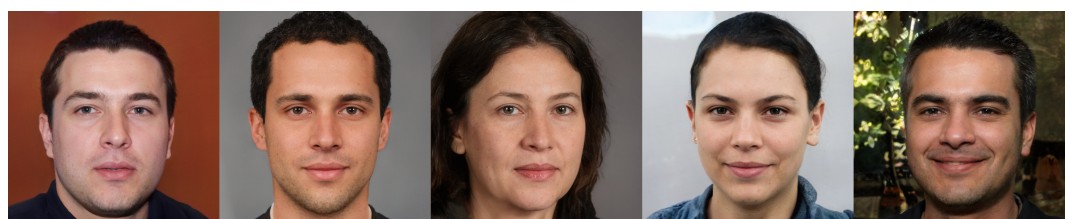

Figure 12: **Reference images.** Reference images used in the video shown in Fig 10 and Fig. 11.

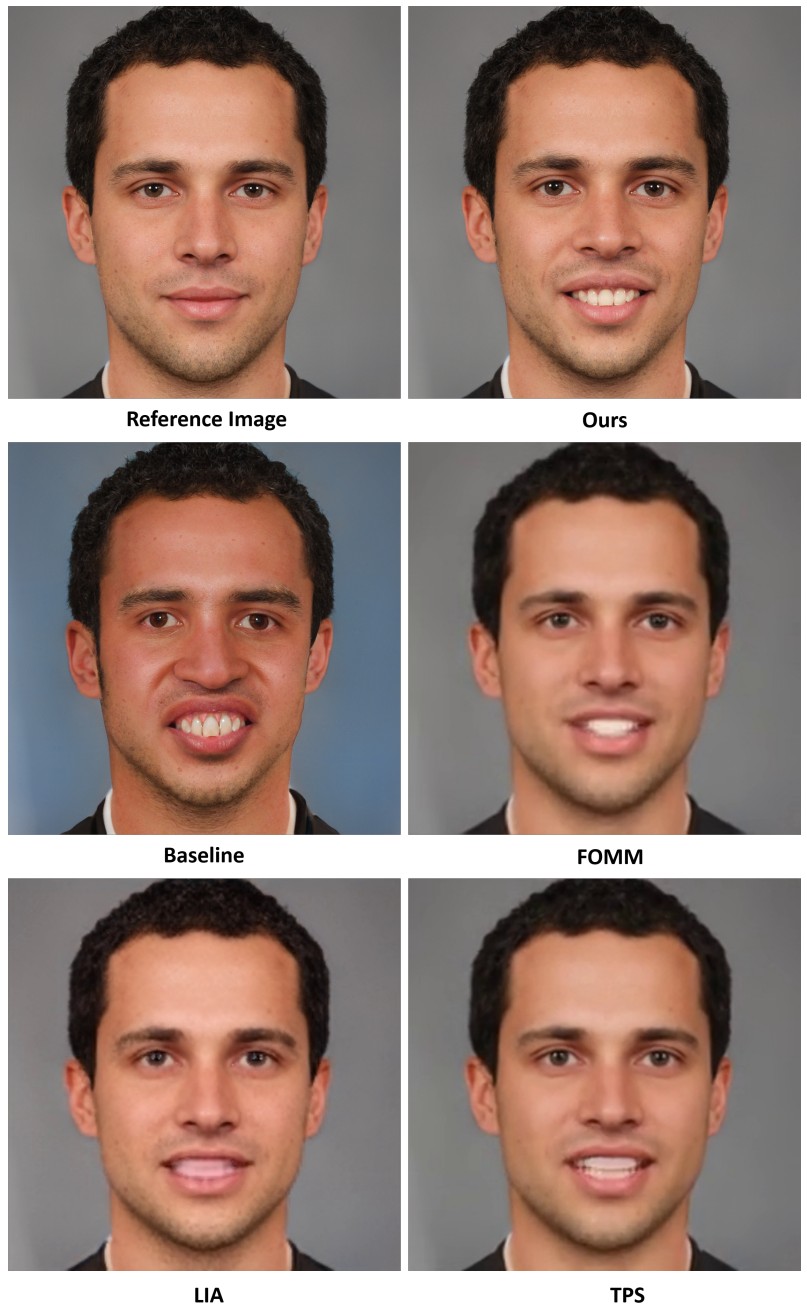

Figure 13: **Visual quality.** Zoomed-in version of images shown in Fig. 10 (fourth row).

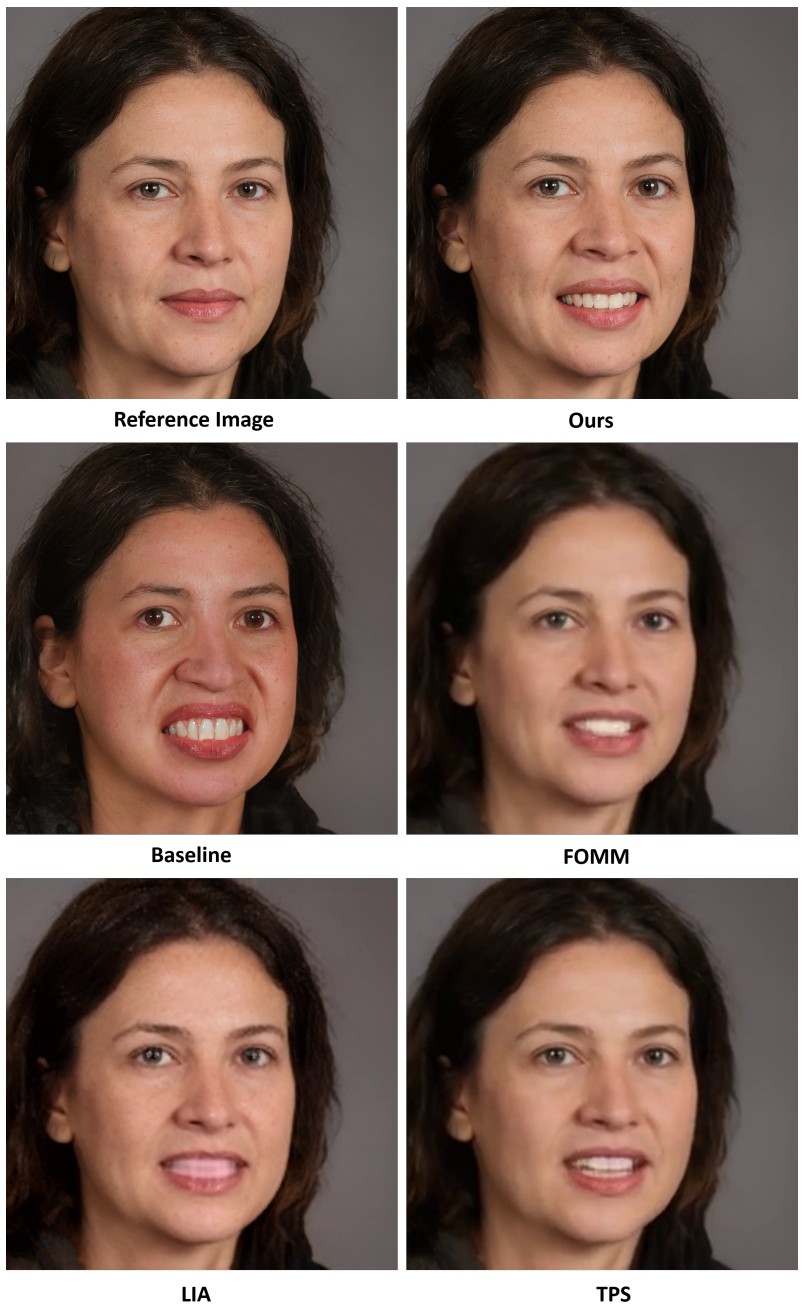

Figure 14: **Visual quality.** Zoomed-in version of images shown in Fig. 10 (sixth row).

