# OpenReview forum: "Video2StyleGAN: Disentangling Local and Global Variations in a Video"
_ICLR.cc/2024/Conference — Submitted to ICLR 2024_

### Official Review · Reviewer_hUSe · 2023-10-29

**Soundness:** 3 good
**Presentation:** 2 fair
**Contribution:** 2 fair
**Rating:** 5
**Confidence:** 4

**Summary:**

The paper proposes Video2Style, a framework focusing on fine-grained human face reenactment. Video2Style digs details like poses,  locations and expressions from driving (or co-driving) video and applies them into the reference videos. The method shows better detail and identity preservation over the previous baselines.

**Strengths:**

Motivation & general ides:
- The paper chooses to explore the fine-grained video manipulation without any 3D information, which seems to be an interesting and meaningful direction.

Method:
- The paper shows some interesting insights into the latent space of StyleGAN3. For example, $W$ space for local variations and $S$ space for global variation.

Results:
- Video2Style shows good control over pose transformation, details (like eyes and mouths), and it produces high-resolution videos.

**Weaknesses:**

Motivation & general ideas:
- The paper claims that a good video manipulation can be achieved by using a fine-grained control over the transformations and facial expressions, but mainly focuses on the reenactment for human faces. The word "video manipulation" is a little ambiguous and not well defined. I would suggest the authors come up with a proper scope of the paper with something related to reenactment.

Method:
- In Section 3.3.2, what is a "masked $W+$ space"? Why does it relate to pose matching and identity preservation.
- In Section 3.3.3, it is known that [BiSeNet](https://github.com/zllrunning/face-parsing.PyTorch) sometimes cannot produce fine-grained masks. Will upsampling these masks with bilinear kernel bring more errors?
- The purpose of using a co-driving  (CD)  video is confusing. What does it do? It is not explained in the paper.

Experiments & results:
- From the results, the texture sticking artifact is very obvious even with a StyleGAN3 generator. Why would this happen given that StyleGAN3 claims it is able to produce anti-aliasing results? I notice that after "+Modified Baseline" (see the boundary of the hair, Ablation of Building Blocks, supplementary website), the texture sticking becomes obvious. Does it relate to this step? And what is the configuration of the StyleGAN3 generator used in the paper?
- With a comparison with previous methods, Video2Style seems to bring some background changes in many examples in the supplementary material. Is there a way to mitigate this artifact?


Writing & presentation:
- The presentation of the paper needs to be improved from my perspective, given the complexity of the proposed method. For example, it is hard to understand Video2StyleGAN pipeline from Figure 2. To be specific, it would be better if the authors could include some figures/examples to demonstrate what each block is doing so that Figure 2 can becomes clearer. Moving some examples from the supplementary material to the main paper seems to be better.
- Also in 3.3.2, the second sentence goes "We observe that applying this change stretches the face area and the eyes, however,
the mouth and nose positions remain unchanged making the output face unrealistic". From the supplementary video "Global Pose Encoding", it is hard to see this artifact from Restyle projection or naive method. The positions of the nose and mouth do change, which makes it hard to see if the proposed method is working or not.

Related work: Some missing related work that might be beneficial:
- [1] Yang, Shuai, et al. "StyleGANEX: StyleGAN-Based Manipulation Beyond Cropped Aligned Faces." arXiv preprint arXiv:2303.06146 (2023).
- [2] Xu, Yiran, Badour AlBahar, and Jia-Bin Huang. "Temporally consistent semantic video editing." European Conference on Computer Vision. Cham: Springer Nature Switzerland, 2022.
- [3] Yao, Xu, et al. "A latent transformer for disentangled face editing in images and videos." Proceedings of the IEEE/CVF international conference on computer vision. 2021.
- [4] Tzaban, Rotem, et al. "Stitch it in time: Gan-based facial editing of real videos." SIGGRAPH Asia 2022 Conference Papers. 2022.

**Questions:**

- In the supplementary website, in "Ablation of Building Blocks" video, what is the "Modified Baseline"?
- Why do you choose StyleGAN3 as your generator if the results presented in the paper still have some texture sticking artifacts? Will use a StyleGAN2 generator better since it is known for better a disentanglement for different semantics in practice?

**Details Of Ethics Concerns:**

The paper could be used for human identity manipulation and possibly bring some misinformation.

---

> ### Author Response · Authors · 2023-11-22
>
> We thank the reviewer for the constructive comments and suggestions on our paper. We address each of the concerns as follows:
>
> 1.**The word "video manipulation" is a little ambiguous.**
>
> Our method does not propose a new generator for videos. We use the pre-trained SG3 generator trained on images. So, technically, it is designed to perform image manipulation. In our work, we combine different latent spaces of StyleGAN3 ($W+$, $S$, $F_f$, and filter weights) and hierarchically combine them in a ‘non-trivial’ manner. This enables us to effectively control the local and global features in a reenacted video, and hence, we can perform ‘’video manipulation’’. We will make it more clear in a subsequent revision.
>
> 2.**Masked $W+$ Space**
>
> In the StyleGAN domain, w/w+ latent codes are input to a number of layers at different resolutions. Each layer controls the characteristics of the output image. In general, the earlier layers are responsible for the coarse features, and the later ones are responsible for finer details. Works like StyleFlow, PSP, etc, show that the identity-preserving edits can be made in the early layers of the StyleGAN. Hence, we ‘mask’ the latter layers so that there is no significant change in the style of the image and only the coarse semantic attributes, in our case, the pose is manipulated.
>
> 3.**BiSeNet Mask upsampling and significance of co-driving video.**
>
> While the upsampling of the masks may bring in more errors, we actually do not need accurate masks to identify the layers responsible for a given attribute. The masks can be coarse, and we can identify the layers that have maximum overlap in terms of IOU scores. In order to show that the rotation and translation motion in the video can be inputted from a different video and hence is disentangled, we use co-driving video to extract this information and apply it to our final videos.
>
> 4.**Texture sticking artifact, background changes, and the generator used.**
>
> Yes, SG3 sometimes shows jitters in the background compared to non-StyleGAN methods, i.e., FOMM and TPS; we would like to argue these methods are keypoint-based and are designed to explicitly avoid jittering. In contrast, our method does not require keypoints to compute the motion, so some jitters are expected with complex backgrounds. However, it is relatively simple to remove background artifacts by using the method in Local Facial Feature Encoding of the main paper on the background in addition to mouth, nose, and eyes. Instead of modeling the local motion, the corresponding $s$ activations of the background feature maps can be suppressed, hence avoiding motions in the background. In general, the artifacts observed in the video stem from the StyleGAN3 generator. It's crucial to note that this particular aspect of our video quality is subject to enhancement with improvements in StyleGAN-based generators. We use the R configuration of StyleGAN3.
>
> 5.**SG2 vs SG3**
>
> A notable distinction between SG3 and SG2 lies in SG3's ability to reposition the head, a feature crucial for a video generation pipeline. This capability proves beneficial in creating realistic motions for talking heads, as demonstrated in our supplementary video. While we acknowledge the presence of certain artifacts inherited from the SG3 generator, we maintain that our method attains commendable video generation quality at a resolution of 1024x1024. This is particularly noteworthy as many alternative methods primarily train on videos and are designed to support a lower 256x256 resolution.
> We recognize the existing artifacts and attribute them to the current state of the SG3 generator. However, we are optimistic that with the ongoing advancements in GAN generators, especially continuous improvements in terms of disentanglement, our method will progressively overcome these artifacts and further enhance its overall performance.
>
> 6.**What is the modified baseline?**
>
> The modified baseline method is only applying the edits to layers 3 - 7 of the w+ space. This is explained in section 3.3 of the paper. We will mention it more explicitly.
>
> 7.**Deformations in the baselines.**
>
> In the example shown in the supplementary video, specifically, the head shape is changed. Notice in the baseline, it creates the unnatural bloating of the head region or increases the head size, as in the example in the first row. In comparison, our method maintains the features of the head while performing the pose change.
>
> 8.**Related works, Figure 2 and moving some parts from the supplement.**
>
> Thanks for pointing out additional related works. We included the related works in the paper. We will improve the figure and move the relevant discussion from the supplement to the main paper.

---

### Official Review · Reviewer_7M9K · 2023-10-30

**Soundness:** 3 good
**Presentation:** 3 good
**Contribution:** 2 fair
**Rating:** 5
**Confidence:** 4

**Summary:**

The paper presents a method for human portrait video generation and manipulation using StyleGAN3 for a target character image using source videos. They encode global (pose) and local (expressions) variations to manipulate a pre-trained SG3's W and S feature spaces, respectively and SG3's Fourier features for better pose control. As a result, co-diving where pose from one video and expressions from another video can be combined thanks to the proposed disentanglement. Several results and comparisons are presented.

**Strengths:**

- The overall strength of the work is hierarchical framework that enables local (facial semantic) and global pose control.
- The paper is generally written well and provides several comparisons against SOTA, where compared image-to-image the method outperforms several SOTA methods wrt to identify preservation and image reconstruction metrics.

**Weaknesses:**

- The main limitation of the work is inconsistent local facial deformations. S-space expression deformations seem to perturb the space spatially inconsistently. Is this an artifact of SG3 or the method? Would you have the same artifact with SG2?
- According to the user study conduced (Appendix N), the results of motion/expressions were accepted ~50% of the time. I am not sure if this a strong indicator of success.
- Only a single method ablation (baseline +  Local Facial Feature Encoding) is provided. There is no discussion as to why this ablation was conduced. Thus, the method analysis seems incomplete.

**Questions:**

- While transferring the expressions from source video, do you normalize the images to remove pose (say using landmarks), s.t. only expression gets transferred? It may help with non-physical facial features?
- Can the authors consider a method similar to Pick-a-Pic, Kirstain et al. '23 to understand if temporal consistency of facial features is better accepted for the current method over SOTA?
- Please add a result where only pose change takes place in target video, to help understand if local deformations such as non-physical nose shape change still occurs?

---

> ### Author Response · Authors · 2023-11-22
>
> We thank the reviewer for the constructive comments and suggestions on our paper. We address each of the concerns as follows:
>
> 1.**SG2 vs SG3**
>
> A notable distinction between SG3 and SG2 lies in SG3's ability to reposition the head, a feature crucial for a video generation pipeline. This capability proves beneficial in creating realistic motions for talking heads, as demonstrated in our supplementary video. While we acknowledge the presence of certain artifacts inherited from the SG3 generator, we maintain that our method attains commendable video generation quality at a resolution of 1024x1024. This is particularly noteworthy as many alternative methods primarily train on videos and are designed to support a lower 256x256 resolution.
> We recognize the existing artifacts and attribute them to the current state of the SG3 generator. However, we are optimistic that with the ongoing advancements in GAN generators, especially continuous improvements in terms of disentanglement, our method will progressively overcome these artifacts and further enhance its overall performance.
>
>
>
> 2.**Explanation of user study and ablation.**
>
> In the supplementary material, we conduct two distinct user studies. The first study assesses the reconstruction quality of identity, and our method is preferred approximately **63**% of the time. This outcome underscores the paper's contribution to generating high-quality videos at a superior resolution. The second user study focuses on evaluating the motion in the video, revealing that our method demonstrates comparable performance to competing methods in this aspect.
> To ensure the robustness of our approach and to affirm that a naive baseline does not yield high-fidelity and temporally consistent videos, we perform an ablation experiment. Specifically, we evaluate the Local Feature Encoding in conjunction with the baseline. The rationale behind this ablation is to demonstrate the contribution of Local Feature Encoding to temporal smoothness and keypoint accuracy.
>
> 3.**Normalizing the images for expression changes.**
>
> Yes, we normalize the images before computing the expression edits. We do this by standard landmark detection and cropping using the code in the StyleGAN repo.
>
> 4.**Pick-a-Pic score for the videos.**
>
> Unfortunately, there are currently no widely adopted large-scale video evaluation methods comparable to the one mentioned for assessing temporal consistency. The reviewer proposed a promising avenue for future work that could greatly benefit the community. In response to this suggestion, we already computed a temporal metric, FVD (Fréchet Video Distance), for our videos, and the results are presented in **Table 4 of the supplementary material**. In this comparison with competing methods, it's important to note that due to the smooth nature of the GAN latent space we employ, there are no discernible jumps or flickering artifacts in the video. Consequently, the FVD metric aligns closely with methods that are trained explicitly on video data.
>
> 5.**Pose change results.**
>
> We have incorporated a video on the **supplementary webpage**, showcasing various poses derived from distinct frames alongside the corresponding face reenactment videos. It is essential to observe that our method demonstrates an ability to disentangle pose information with localized edits.

---

> > ### Comment · Reviewer_7M9K · 2023-11-22
> > **Small scale non-physical movements of rigid face attributes.**
> >
> > Thanks for your comments. Personally, the small-scale non-physical movements, for example the nose change it size as the person talks, even when only pose changes suggest that the local space disentanglement is still off.
> >
> > If we were to calculate optical-flow between the frames of real videos vs generative videos, this local deformation change would be clearer. Also given that the emphasis of the paper is high-resolution video generation -- as we go high-res, local features are perceptually more visible. As a result, I am not able to recommend.

---

> > > ### Author Response · Authors · 2023-11-22
> > >
> > > We acknowledge that there is room for improvement. However, we would like to emphasize that the criterion for evaluation should be the improvement over previous work. We have demonstrated clear improvements over the previous methods in this work. This should make the work publishable at a high level.

---

### Official Review · Reviewer_uEde · 2023-10-31

**Soundness:** 2 fair
**Presentation:** 2 fair
**Contribution:** 2 fair
**Rating:** 3
**Confidence:** 4

**Summary:**

This paper explores the use of a pre-trained StyleGAN generator for image editing, particularly focusing on facial manipulation. While StyleGAN has proven effective for controlling various facial attributes like age and expression in images, the researchers identified limitations when applying it to video editing. The key challenge appears to be the need for precise and disentangled control over factors such as face location, pose, and local facial expressions. To address this, the researchers propose a novel approach that involves working across multiple latent spaces, including positional, W+, and S spaces, and combining optimization results. They introduce Video2StyleGAN, a method capable of reenacting the local and global features, expressions, and locations from a driving video onto a target image's identity. The outcome is the generation of high-quality videos at an impressive 10242 resolution without requiring specific training on video data.

**Strengths:**

The simultaneous working across multiple latent spaces, as described in the approach, represents an innovative and promising direction in the field of image and video editing.

**Weaknesses:**

1.	The motivation is unfounded, as StyleHEAT and other methods using StyleGAN naturally support videos.
2.	MEAD is primarily an emotion-focused dataset, making it unsuitable as an experimental dataset for reenactment. The authors should conduct a fair comparison on more datasets and with more recent methods.
3.	The  generated results have limitations, including low mouth shape precision, low facial expression control accuracy, and the ability to control only frontal poses.

**Questions:**

1.	Could you provide more explanation and evidence for the motivation?
2.	Why not evaluate on a dataset that is typically used in reenactment?
3.	Why not compare with recent SOTAS, e.g. StyleHEAT?
4.	Can you provide results of pose driving along different axes?
5.	Why do the teeth in the supplementary material videos appear to have significant artifacts?

---

> ### Author Response · Authors · 2023-11-22
>
> We thank the reviewer for the constructive comments and suggestions on our paper. We address each of the concerns as follows:
>
> 1.**The motivation of the work.**
>
> The paper consistently presents a clear and well-defined motivation supported by evidence both in the main text and the supplementary materials. Firstly, our objective is distinct: we address the challenge of high-quality video generation and face reenactment at a resolution of 1024x1024 using a GAN without relying on any video training. Notably, competing works typically achieve generation at a lower resolution (256x256) and are trained on video data. Second, our focus extends beyond resolution; we aim to enhance fidelity in the results at this higher resolution. The paper demonstrates substantial improvements in reconstruction quality and the preservation of details, accompanied by comparable performance in motion and expression transfer, as evidenced in Tables 1 and 2 in the main paper. While we acknowledge that further research could enhance motion and expression transfer, our overall approach outperforms prior art.
> Third, we emphasize the importance of a facial reenactment system that provides control over both local and global features in video generation. Our StyleGAN3-based framework, as illustrated in the paper and supplementary video, effectively disentangles various local and global properties, including expressions in a video, and independently controls them. This capability is not demonstrated in previous video editing works trained on videos, let alone in a network solely trained on images.
> Finally, given the absence of high-quality video datasets and other network limitations, our rationale for choosing StyleGAN3 is based on the need for solutions that use a smooth latent space to model videos. The generator is not inherently video-based; the innovation lies in our exploration and combination of regularized latent spaces to make it suitable for video editing. To achieve this, we meticulously examine different latent spaces of StyleGAN3, including $W+$, $S$, $F_f$, and filter weights, and we are the first to hierarchically combine them in a 'non-trivial' manner.
>
>
> 2.**StyleHEAT and other methods naturally support videos, and why did we not compare with StyleHEAT?**
>
> We firmly disagree with the reviewer's assessment. StyleHeat utilizes the pretrained latent space of StyleGAN2 for video generation. In our paper, we have conducted a thorough comparison of our results with the competing methods. It is important to highlight that StyleGAN3 offers support for head repositioning, a capability that StyleGAN2 lacks. Our method is compared specifically with StyleHEAT, and we have already presented the comparison results for this method in **Supplement Table 3, Figure 8, and with a video-based evaluation** accessible on the attached webpage.
>
>
> 3.**Results of Pose changes.**
>
> We have incorporated a video on the **supplementary webpage**, showcasing various poses derived from distinct frames alongside the corresponding face reenactment videos. It is essential to observe that our method demonstrates an ability to disentangle pose information with localized edits.
>
>
> 4.**Other datasets?**
>
> We want to emphasize that our approach is not trained on extensive video datasets. Our comparison with the provided dataset remains fair, as all the methods considered are impartial to the training samples within the MEAD dataset. This dataset encompasses diverse actors expressing emotions with varying intensities, enabling the capture of micro-expressions and finer details. This diversity is crucial for evaluating high-resolution video generation, such as the one we propose. Consequently, the MEAD dataset serves as a robust test set for all methods, including ours.
>
> 5.**Artifacts in the video (teeth).**
>
> The artifacts observed in the video stem from the StyleGAN3 generator. It's crucial to note that this particular aspect of our video quality is subject to enhancement with improvements in StyleGAN-based generators.

---

> ### Comment · Reviewer_uEde · 2023-12-03
> **Comments After Rebuttal**
>
> While the author claims to present a method for video editing, a significant portion of the experimental results leans towards reenactment aspects. If positioned as a reenactment method, it fails to comprehensively demonstrate pose control along the row, yaw, and pitch axes. Upon closer examination of the videos, it is evident that the accuracy of single-frame expression driving is relatively low. Regardless of the research direction, the experiments conducted with this method are incomplete, and the comparisons are inadequately presented.
> The author has not fully addressed my concerns. Regarding the dataset comparison issue, shuffling the image sequence in video datasets can easily address the stated reason for not training on video datasets. MEAD fails to showcase the diversity of pose and expression. Throughout the entire paper, the author neglects to demonstrate pose control along other axes, which is a perplexing omission.

---

### Official Review · Reviewer_1vN3 · 2023-10-31

**Soundness:** 4 excellent
**Presentation:** 3 good
**Contribution:** 4 excellent
**Rating:** 8
**Confidence:** 4

**Summary:**

Editing images with a pre-trained generative model has been extensively explored. However, when it comes to applying controllable generation to video editing, several challenges arise. The authors of the paper claim that the main obstacle is the lack of precise control over face location, pose, and local expressions. They illustrate that utilizing a pre-trained StyleGAN enables the achievement of fine-grained control by concurrently traversing multiple latent spaces (positional, W+, and S spaces) and combining the results of optimization. Consequently, the authors introduce Video2StyleGAN, a method that employs a target image and driving video(s) to reproduce local and global facial features from the driving video onto the identity of the target image, generating a high-resolution video. Extensive experiments have shown the effectiveness of the approach in difficult scenarios.

**Strengths:**

- The novel utilization of the invariance property of StyleGAN3 is noteworthy.
- Upon checking the supplementary material, extensive experiments have demonstrated the applicability of the approach.
- Despite requiring optimization, the speed appears to be sufficiently fast (1.5 frames/sec).
- The hierarchical design of each component allows for controllability over different properties, which are also well-described.
- The approach achieves competitive results with previous methodologies.

**Weaknesses:**

- At first glance, Figure 2 appears unclear; enhancing its clarity would improve the presentation of your methodology.
- Unfortunately, the approach still requires some fine-tuning/optimization. Have you considered any approaches to minimize the need for optimization?

**Questions:**

- Extending the approach to latent diffusion models and observing the results, as mentioned in the limitations section, would be intriguing. How do you plan to identify counterparts to these properties in GANs within diffusion models?
- I'm curious about your strategy for determining the use of 3-7 layers. Could you elaborate on how you arrived at this decision and, if possible, share relevant results?
- You noted that optimization on w is specifically applied to the first 8 layers. Could you provide more details on the reasoning behind this choice?

---

> ### Author Response · Authors · 2023-11-22
>
> We thank the reviewer for the constructive comments and suggestions on our paper. We address each of the concerns as follows:
>
> 1.**Figure is unclear**
>
> Thanks for the suggestion. We will improve the figure in an eventual final version of the paper.
>
> 2.**The method uses fine-tuning/optimization. Any approach to minimize it?**
>
> The most time-consuming phase in our Video2StyleGAN method is the GLOBAL POSE ENCODING step, which encompasses pose optimization utilizing the w space of StyleGAN3. Although this component of the method relies on optimization, a viable alternative could involve adopting a learned pose regressor, akin to approaches seen in the StyleGAN2 domain, such as StyleFlow. Exploring this alternative is on our agenda for future research.
>
> 3.**Counterparts of our components to Diffusion models**
>
> One can compare the counterparts of our method with h-space and the skip connections in the Diffusion domain. For example, the” Training-free Style Transfer Emerges from h-space in Diffusion models” paper discovers that DMs inherently have disentangled representations for content and style of the resulting images: h-space contains the content, and the skip connections convey the style.
>
> 4.**The strategy of using 3 - 7 layers and pose edit applied to only the first 8 layers.**
>
> In the StyleGAN domain, it is a common practice to utilize a subset of the w+ space for identity-preserving edits. Numerous works in the StyleGAN domain, such as StyleFlow and StyleCLIP, employ various strategies to identify the layers responsible for these edits. In our approach, we determined the layers that, after the edit, do not alter the identity and style of the person. This determination is achieved by leveraging ArcFace embeddings and comparing the original image before and after the edits introduced by these layers.

---

### Meta-Review · Area_Chair_53sa · 2023-12-03

**Metareview:**

Summary: The paper explores a technique for applying a target image's local and global features to a driving video, without the need for training on video data. The motivation behind this work is to address the limitations of StyleGAN in achieving precise and disentangled control of local features during video editing. Strengths: The paper introduces an innovative hierarchical embedding method that operates across multiple latent spaces (positional, S, and w+ spaces). It also demonstrates the ability to generate 1024x resolution video without training on video data. Weaknesses: While the innovation is appreciable, it is rather specific to facial reenactment, despite the paper's broader objective of video manipulation. The results do not appear strong enough, displaying obvious sticking artifacts. Additionally, several outstanding questions from reviewers remain, and the rebuttal has not been convincing enough for me to downplay the reviewers’ concerns.

**Justification For Why Not Higher Score:**

The paper received 2x below acceptance, 1x reject and 1x accept. The method is interesting. Its scope should be revisited, results remain questionable.

**Justification For Why Not Lower Score:**

N/A

---

### Decision · Program_Chairs · 2024-01-16

Reject